# *Cannabis sativa* L. Leaf Oil Displays Cardiovascular Protective Effects in Hypertensive Rats

**DOI:** 10.3390/ijms26051897

**Published:** 2025-02-22

**Authors:** Juthamas Khamseekaew, Monchai Duangjinda, Putcharawipa Maneesai, Chanon Labjit, Siwayu Rattanakanokchai, Sudarat Rongpan, Poungrat Pakdeechote, Prapassorn Potue

**Affiliations:** 1Department of Physiology, Faculty of Medicine, Khon Kaen University, Khon Kaen 40002, Thailand; juthakh@kku.ac.th (J.K.); putcma@kku.ac.th (P.M.); sudarat.ro@kkumail.com (S.R.); ppoung@kku.ac.th (P.P.); 2Department of Animal Science, Faculty of Agriculture, Khon Kaen University, Khon Kaen 40002, Thailand; monchai@kku.ac.th; 3Department of Horticulture, Faculty of Agriculture, Khon Kaen University, Khon Kaen 40002, Thailand; lchano@kku.ac.th; 4Faculty of Veterinary Medicine, Khon Kaen University, Khon Kaen 40002, Thailand; siwara@kku.ac.th; 5Department of Physiology, Faculty of Medicine, Princess of Naradhiwas University, Narathiwat 96000, Thailand

**Keywords:** *Cannabis sativa* L. leaf oil, hemp leaf oil, cardiovascular function, L-NAME-induced hypertensive rats

## Abstract

Hemp (*Cannabis sativa* L.) leaf oil (HLO) contains several bioactive compounds such as phenolics, flavonoids, and quercetin. However, the effects of HLO on hypertensive conditions have not yet been investigated. This study investigated the cardiovascular protective effects of HLO in a nitric oxide (NO) synthase inhibitor-induced hypertensive rat model. Five weeks of HLO administration significantly prevented blood pressure elevation, improved cardiac function, and mitigated cardiac hypertrophy. Furthermore, HLO ameliorated vascular dysfunction by reducing sympathetic nerve stimulation-induced vasoconstriction, increasing endothelium-dependent vasorelaxation, as well as decreasing vascular wall thickness and vascular smooth muscle cell proliferation. HLO inhibited renin–angiotensin system (RAS) activation and downregulated angiotensin II type 1 (AT1) receptor and NADPH oxidase expression. Additionally, HLO normalized the circulating NO metabolites, decreased oxidative stress, and enhanced antioxidant status. These findings suggest that HLO protects against cardiovascular dysfunction and preserves its morphology. The mechanism of action might involve the suppression of RAS overactivity and oxidative stress through the Ang II/AT1 receptor/NOX2 pathway in NO-deficient hypertension.

## 1. Introduction

Hemp (*Cannabis sativa* L.) is a plant in the Cannabaceae family. It has become increasingly popular worldwide because of its pharmacological, recreational, and medical benefits [1]. The food industry has developed hemp bioactive substances as functional ingredients in various foods including pasta, snacks, bread, and cookies [2,3]. The major bioactive components, terpenes, cannabinoids, phenolic acids, and flavonoids isolated in hemp leaves have been demonstrated [4,5]. These constituents might mediate and support their beneficial effects such as antidepressant, anti-inflammatory, antioxidant, and antiproliferative effects [5]. It has been reported that *Cannabis Sativa* L. leaf extract increases glucose utilization, reduces oxidative stress and cholinergic impairment, and modulates purinergic and gluconeogenic functions in the cerebral tissues of rats [6]. The pharmacological studies show that hemp leaf oil (HLO) exhibits a marked cytotoxicity regarding cancer cells, especially cholangiocarcinoma cells [4]. The potential effects of hemp leaf extract on incision wound healing have been demonstrated in rats [7]. It has been proven that *Cannabis sativa* L. displays low levels of psychoactive cannabinoids and holds an important source of flavonoids [8]. A previous study reported that *Cannabis sativa* L., the non-psychoactive residual biomass (stems and leaves), exhibits total antioxidant capacity and total phenolic content [9]. *Cannabis sativa* L. and its bioactive compounds have been increasingly studied for their potential cardiovascular effects. However, its mechanism of action is complex and has a multidirectional influence on the cardiovascular system. Nevertheless, limited research exists regarding the blood pressure and cardiovascular effects of HLO.

Hypertension is a chronic condition linked to cardiovascular and vital organ impairment. Primary or essential hypertension has an unidentified etiology but may be linked to vascular endothelial dysfunction. This abnormality is primarily caused by reductions in nitric oxide production and bioavailability [10]. An animal model of hypertension has been created to mimic the reduction in endothelial function by inhibiting nitric oxide generation using N(ω)-nitro-L-arginine methyl ester (L-NAME) [11]. Several studies uncovered that large and small arteries isolated from rodents treated with L-NAME lack endothelial function to mediate vasorelaxation [12,13,14]. Furthermore, L-NAME therapy augments pressor responses to sympathetic nerve stimulation (SNS), hence sustaining elevated blood pressure in this hypertension paradigm [15,16]. Impairment of cardiac performance because of hemodynamic overload induced by nitric oxide deficiency has been confirmed in rodents [17,18]. Left ventricular-aortic thickness and fibrosis are observed in NO depletion-induced hypertensive rats [19,20]. These morphological changes are mainly caused by hemodynamic overload and activation of the renin–angiotensin system (RAS) downstream.

Under physiological settings, the RAS is largely responsible for the long-term regulation of blood pressure. Nonetheless, the over-activation of this system significantly contributes to the etiology of cardiovascular hypertrophy and remodeling [21]. Angiotensin II (Ang II) is synthesized from angiotensin I by an angiotensin-converting enzyme (ACE) [22]. Once it binds to the Ang II type 1 (AT1) receptor, it becomes a potent vasoconstrictor and hypertrophic agent [23]. Furthermore, Ang II-mediated cardiac hypertrophic response is dependent on the redox-signaling pathway [24]. Evidence supports that Ang II binds to AT1 receptors, triggering the activation of nicotinamide adenine dinucleotide phosphate (NADPH) oxidase, which subsequently generates superoxide [25]. The authors showed that blocking vascular NADPH oxidase isoforms can reduce superoxide production and hypertension induced by Ang II. Recently, the overexpression of NADPH oxidase in cardiac and aortic tissue has been reported to be associated with oxidative stress in hypertensive rats [26,27].

Due to the limited information available on the impact of HLO on NO-deficient hypertension, particularly concerning cardiovascular alterations, the aim of the present study was to investigate the effect of HLO on blood pressure and cardiovascular changes in Sprague Dawley rats treated with L-NAME. Additionally, we aimed to investigate the possible mechanisms underlying these effects.

## 2. Results

### 2.1. Bioactive Compounds Found in HLO

The total phenolic compounds and flavonoid contents were found in HLO, measured using spectrophotometer methods. The quercetin concentration in HLO was also detected through high performance liquid chromatography (HPLC) (Table 1).

### 2.2. HLO Prevents the Gradual Increase in Systolic Blood Pressure (SBP) in Hypertensive Rats

During the experimental period, rats treated with L-NAME developed an increase in SBP (184.75 ± 4.06 mmHg) compared to that of the control rats (116.38 ± 1.02 mmHg, *p* < 0.05). L-NAME rats treated with HLO at doses of 1.0 and 1.5 mL/kg/day exhibited a significant reduction in the gradual increase in SBP (151.38 ± 3.34 and 147.38 ± 3.04 mmHg, respectively) compared to that in the HT group (*p* < 0.05) (Figure 1). The findings underscore the potential role of HLO in blood pressure regulation, as it effectively prevented the elevated blood pressure observed in hypertensive rats.

### 2.3. HLO Decreases High Blood Pressure in Hypertensive Rats

At the end of the experiment, rats in the hypertension group showed a significant increase in SBP, diastolic blood pressure (DBP), and mean arterial pressure (MAP) compared to that of control (*p* < 0.05). This high blood pressure was reduced with daily HLO supplementation (*p* < 0.05) (Table 2). The data highlight the beneficial effects of HLO on hypertension caused by NO deficiency.

### 2.4. Effects of HLO on Body Weight and Organ Weight in Hypertensive Rats

No significant differences in body weight were observed among the rats in each group. However, the heart weight to body weight ratio (HW/BW), left ventricular weight (LVW), and LVW/BW ratio were significantly increased in the HT group compared to the results for the control group (*p* < 0.05). These increases in the heart weight parameters suggest the development of hypertrophic changes in the heart due to hypertension. Treatment with HLO improved the LVW/BW ratio in hypertensive rats (*p* < 0.05) (Table 3). This indicates that HLO may have a protective effect against cardiac hypertrophy associated with hypertension.

### 2.5. HLO Improves the Alteration of Cardiac Function Parameters in Hypertensive Rats

The assessment of cardiac function by echocardiography revealed a significant increase in the left ventricular posterior wall thickness during diastole (LVPWd) and in the interventricular septum during diastole (IVSd), along with a decrease in left ventricular internal dimension at end-diastole (LVIDd), end-diastolic volume (EDV), ejection fraction (EF), stroke volume (SV), and fractional shortening (FS) in HT rats compared to the results for the control rats (*p* < 0.05), suggesting a decline in the heart’s ability to contract and pump blood effectively. Treatment with HLO significantly improved cardiac function parameters, including LVPWd, EF, SV, and FS, compared to untreated hypertensive rats (*p* < 0.05) (Table 4), suggesting that HLO has a beneficial effect on cardiac function in hypertensive condition.

### 2.6. HLO Improves Vascular Responses to Sympathetic Nerve Stimulation and Endothelial Function in Hypertensive Rats

The contractile responses to sympathetic nerve stimulation were assessed using isolated mesenteric vascular beds. Hypertensive rats exhibited significantly greater contractile responses to electrical field stimulation (EFS) compared to that of the control rats (*p* < 0.05). However, HLO treatment suppressed the EFS response in hypertensive rats (*p* < 0.05) (Figure 2A), indicating that HLO may have a modulating effect on sympathetic tone and vascular reactivity in hypertension. No significant differences were observed in the response to exogenous norepinephrine (NE) among the groups (*p* > 0.05) (Figure 2B), suggesting that the effects of HLO are specific to the modulation of the sympathetic nervous system at the pre-synaptic site.

Furthermore, endothelial function was evaluated in both the mesenteric vascular beds and aortic rings. Hypertensive rats showed a significant reduction in the perfusion pressure of mesenteric vascular beds and impaired relaxation of the aortic rings in response to acetylcholine (ACh) compared to the results for the control rats (*p* < 0.05), indicating endothelial dysfunction and reduced NO-mediated vasodilation. Treatment with HLO for five weeks significantly improved the relaxation response to ACh in hypertensive rats (*p* < 0.05) (Figure 2C,E). No significant differences were observed in the response to exogenous sodium nitroprusside (SNP) among the groups (*p* > 0.05), indicating that HLO treatment did not affect the response of the smooth muscle cells to direct vasodilators (Figure 2D,F).

### 2.7. HLO Improves Cardiac Hypertrophy in Hypertensive Rats

LV hypertrophy was observed in the hypertension group, as shown in Figure 3. Receiving L-NAME for five weeks resulted in a significant increase in LV wall thickness and cross-sectional area (CSA) compared to the results for the control group (*p* < 0.05). Notably, HLO treatment prevented these alteration in LV morphology when compared to the results for the untreated hypertensive rats (*p* < 0.05). These observations reveal that HLO mitigates the structural changes in the heart associated with hypertension.

### 2.8. HLO Improves the Alteration of Vascular Morphology in Hypertensive Rats

As shown in Figure 4, significant changes in the structure of the thoracic aorta were observed in hypertensive rats, including increases in wall thickness, cross-sectional area (CSA), wall-to-lumen ratio, and vascular smooth muscle cell (VSMC) proliferation, compared to the results for the control group (*p* < 0.05). No differences were found in the luminal diameter across the groups. Treatment with HLO for five weeks resulted in improvements in vascular hypertrophy and reduced VSMC proliferation in hypertensive rats compared to the results for the untreated group (*p* < 0.05), suggesting that HLO plays a protective role in reducing vascular remodeling and smooth muscle cell proliferation in hypertension.

### 2.9. HLO Modulates RAS in Hypertensive Rats

A significant elevation in serum ACE activity, plasma Ang II levels, and AT1 receptor protein overexpression in both aorta and cardiac tissues were found in rats treated with L-NAME for five weeks compared to the results for the control (*p* < 0.05). Supplementation with HLO at doses of 1.0 and 1.5 mL/kg/day significantly reduced RAS overactivity in hypertensive rats as compared to the results for the untreated hypertensive group (*p* < 0.05) (Figure 5). These results indicate that HLO may exert a modulatory effect on the RAS.

### 2.10. HLO Improves NADPH Oxidase Expression in Both Aortic and Cardiac Tissues of Hypertensive Rats

Rats treated with L-NAME exhibited a significant increase in the protein expression of gp91^phox^ in both aorta and cardiac tissues compared to the results for the control rats (*p* < 0.05). However, supplementation with HLO at doses of 1.0 and 1.5 mL/kg/day significantly reduced the overexpression of gp91^phox^ protein in hypertensive rats when compared to that in the untreated hypertensive group (*p* < 0.05) (Figure 6). HLO may exert antioxidative effects by attenuating the overexpression of gp91^phox^, a key subunit of NADPH oxidase, which is involved in reactive oxygen species (ROS) generation.

### 2.11. HLO Improves NO Level and Oxidative Status in Hypertensive Rats

The HT group exhibited a significantly lower level of NO metabolites compared to the normotensive group (*p* < 0.05), but HLO treatment restored these levels (*p* < 0.05). The oxidative stress markers, including superoxide production in both aorta and cardiac tissues, as well as plasma and cardiac tissue malondialdehyde (MDA) levels, were significantly higher in hypertensive rats compared to the levels for the control rats (*p* < 0.05). Furthermore, catalase activity in both serum and cardiac tissues was significantly reduced in hypertensive rats when compared to that for the control rats (*p* < 0.05). HLO at doses of 1.0 and 1.5 mL/kg/day significantly improved these oxidative stress parameters in hypertensive rats in comparison to those for the untreated hypertensive group (*p* < 0.05) (Table 5). The data highlight the fact that HLO may exert antioxidative effects that help to restore NO levels and improve oxidative stress markers in hypertensive rats.

## 3. Discussion

This study demonstrates the cardiovascular-protective effects of HLO in chronic NO-deficient rats. The HLO used in this study contains phenolic compounds, including total phenolic compounds, total flavonoids, and quercetin. HLO partially prevents elevated blood pressure induced by a nitric oxide synthase inhibitor in rats. It also improves the echocardiographic parameters LVPWd, SV, FS, and EF in rats that received L-NAME. The hypertensive group exhibited enhanced SNS-mediated vasoconstriction and vascular endothelial dysfunction. These vascular dysfunctions were alleviated by HLO supplementation. LV hypertrophy and increased aortic thickness were found in the hypertensive group but were not seen in the HLO-treated group. The RAS was activated in the HT group, as evidenced by the increased ACE activity and Ang II levels, but these were inhibited in the HT + HLO group. AT1 receptor and NOX2 protein expression were elevated in the HT group, and these protein expressions were downregulated by HLO treatment. The alterations of the biochemical parameters in the circulation and tissue, including NO metabolites, superoxide production, MDA, and catalase, were corrected by HLO treatment.

The animals in the current study developed elevated blood pressure as a result of chronic treatment with the NO synthase inhibitor. Our findings corroborate the results of prior studies indicating that hypertension can arise from the restriction of vascular endothelial function to reduce nitric oxide generation [28,29]. Under physiological conditions, nitric oxide is a vital gas synthesized by vascular endothelial cells that respond to shear stress and vasoactive agents [30]. Thus, administration of NOS inhibitors causes the depletion of NO production and increases vascular tone, total peripheral resistance, and hypertension [31]. High blood pressure induced by NO deficiency in this study is associated with vascular dysfunction in the conduit artery and resistance in the vascular beds. We observed an amplified response to EFS, with no alterations in response to exogenous NE, in the mesenteric vascular bed, indicating pre-junctional enhancement in the hypertensive group. Furthermore, impaired responses to ACh but not SNP in the mesenteric beds and aortic rings were evident in the hypertensive groups, indicating endothelial dysfunction. HLO supplementation improved vascular function and alleviated high blood pressure in a dose-dependent way. The effective antihypertensive dose of HLO in this study was 1 mL/kg. It is well established that NO is a key factor in controlling vascular diameter. HLO increased NO bioavailability, which might improve endothelial function and suppress sympathetic drive.

It is well established that RAS overactivity and oxidative stress play crucial roles in the long-term regulation of high blood pressure in L-NAME-treated rats [32]. Thus, the role of HLO in the long-term regulation of blood pressure, particularly in relation to RAS activation and oxidative stress, has been revealed. We found that HLO treatment normalized the RAS parameters in hypertensive rats. The activation of RAS activity was seen in NOS inhibitor-treated rats because of a reduction in renal blood flow [33]. ACE inhibition activity of HLO exhibits minimal proof, although the inhibitory impact of peptides extracted from *Cannabis sativa* L. seeds on ACE activity that underlies its antihypertensive effects has been described in spontaneously hypertensive rats [34]. The imbalances of superoxide, MDA contents, catalase activity, and NO metabolites in the circulation and tissues were resolved by HLO treatment. Our findings align with the results of a previous study showing that *Cannabis sativa* L. leaf contains phenolic content and exhibits antioxidant capacity [9]. *Cannabis sativa* L. is a significant source of natural antioxidants, as it decreases oxidative stress in blood and brains of mice [35]. It is possible that the HLO-mediated antihypertensive effects are connected to suppressing the RAS system and removing reactive oxygen species.

LV performance was reduced in rats subjected to L-NAME administration. This LV dysfunction is involved in cardiac hypertrophy that fails to adapt to a high-pressure load. In this study, hypertensive rats displayed increases in LV weight relative to BW, IVSd, IVIDd, IVPWd, LV wall thickness, and LV CSA. Furthermore, hypertensive rats exhibited aortic hypertrophy characterized by increases in aortic CSA, wall thickness, the wall to lumen ratio, and vascular smooth muscle cell proliferation, with no changes in luminal diameter. The hypertrophy process is complex, involving hemodynamic alterations and humoral stimulation [36]. HLO supplementation prevents the development of organ hypertrophy induced by NO depletion. These HLO actions should be relevant to two key mechanisms, as mentioned above. First, HLO lowered blood pressure and then reduced cardiac stress. Second, HLO inhibited the RAS pathway, which is a crucial contributor to accelerating cardiac hypertrophy.

It has been proposed that the RAS system, particularly Ang II/AT1 receptor, is a crucial element in a cellular hypertrophic response [37]. The downstream of Ang II/AT1 receptor/NADPH oxidase to mediate cellular hypertrophy has been strongly indicated [24]. In the hypertensive group, we observed the overexpression of AT1 receptor and NOX2 protein in both the heart and aortic tissues, which was consistent with the overproduction of tissue superoxide. NOX2 or gp91^phox^ predominantly produces superoxide and is mostly found in the heart and aorta [38]. Our results confirm the previous reports that Ang II-induced superoxide production via activation of the gp91^phox^ components of NADPH oxidase [25]. Interestingly, the results revealed that animals given L-NAME + HLO displayed heart and aortic tissue AT1 receptor/gp91^phox^ expression levels comparable to those of normal control rats. It is likely that HLO rectified the cardiovascular hypertrophy in hypertension by inhibiting ACE and reducing AT1 receptor/gp91^phox^. Alternatively, it displays an antioxidative action that reduces reactive oxygen species, as described above.

## 4. Materials and Methods

### 4.1. HLO Preparation

Hemp seeds of the Thailand monoecious hemp variety RPF3, developed by Highland Research and Development Institute, Ministry of Agriculture and Cooperatives, Government of Thailand, were used in this study. This hemp was cultivated in experimental plots at the Cannabis Research Farm, Faculty of Agriculture, Khon Kaen University, Thailand (16.471° N, 102.8115° W). The vegetative stage light duration was 14–16 h. The cultivation conditions included nitrogen fertilization (300 kg/ha) and a density of 6000 seedlings/ha. The leaves were harvested at the early-flowering stage, chopped into small pieces, and dried at 40 °C for 3–5 days before being stored in zippered plastic bags at 25 °C.

The hemp extract was prepared through a two-step solvent extraction process. Initially, the hemp leaf material was soaked in a cellulolytic enzyme solution within a closed container for 7 days to facilitate cell-wall digestion. The aqueous phase was subsequently mixed with an organic solvent (absolute ethanol) at 25 °C. Following this, the extract was dissolved in a mixture of ethanol and water at 80 °C, then subjected to ethanol evaporation under a 50 mbar vacuum for a minimum of 4 h. In the final stage, the extract was combined with food-grade olive oil in a ratio of 1:4 to produce edible HLO.

The quantification of cannabinoids in hemp leaf was determined via the HPLC method using a Shimadzu Prominence-i LC-2030 C 3D Plus system, equipped with a Millipore Sigma Ascentis-C18 Express column (2.7 μm × 150 mm × 3 mm) and a photodiode array detector (PDA). The column was operated with solvent A consisting of HPLC water, 8% (*v*/*v*) methanol, 0.035% (*v*/*v*) formic acid, and 1.8 mM ammonium formate, and solvent B comprised of HPLC acetonitrile, with a flow rate of 0.45 mL/min; the UV detector was set at 220 nm. This resulted in an HLO containing 0.40 mg/g cannabidiol (CBD) and less than 0.02 mg/g tetrahydrocannabinol (THC), compliant with Thailand FDA regulations for food and drink products.

The bioactive compounds of HLO were determined by the Nutraceutical Research and Innovation Laboratory, Chiang Mai University, Chiang Mai, Thailand (Table 1). The total phenolic compounds and total flavonoid contents were measured using an in-house method base on a spectrophotometer. The results were expressed as micrograms of quercetin equivalent per mL extract. The quercetin concentration was detected via the HPLC method using a Shimadzu model LC-20 system equipped with a Rheodyne injector and an Inertsil ODS-C18 column (3.5 μm × 150 mm × 4.6 mm). The column was operated in isocratic mode (100:MeOH) at a flow rate of 1.0 mL/min; the UV detector was set at 360 nm.

### 4.2. Animals and Experimental Protocols

Male Sprague Dawley rats (200–220 g) were sourced from Nomura Siam International Co., Ltd., Bangkok, Thailand. The rats were housed in facility with a heating, ventilation, and air-conditioning system maintaining a temperature of 23 ± 2 °C and a 12 h light–dark cycle. All procedures involving the animals adhered to ethical standards for laboratory research and received approval from the Animal Ethics Committee at Khon Kaen University, Thailand (IACUC-KKU-11/65).

After one week of acclimatization, the rats were randomly assigned to five groups (*n* = 8/each group), including a control group, receiving normal drinking water; a hypertension group (HT), receiving L-NAME 40 mg/kg/day dissolved in drinking water; a hypertension group, receiving L-NAME 40 mg/kg/day dissolved in drinking water, together with a daily oral gavage of HLO 0.5, 1.0, and 1.5 mL/kg/day (HT + HLO 0.5, HT + HLO 1.0 and HT + HLO 1.5) over the five weeks of the experimental period.

### 4.3. Indirect Blood Pressure Measurement in Conscious Rats

Systolic blood pressure (SBP) was monitored one time per week using the noninvasive tail-cuff plethysmography technique, employing CODA software (Kent Scientific., Torrington, CT, USA).

### 4.4. Cardiac Function Assessment

At the end of the five-week experimental period, the rats were anesthetized with thiopental sodium (60 mg/kg, intraperitoneally), and the chest area was prepared by shaving and cleaning. Cardiac assessments were conducted using an echocardiogram (Mindray Vetus 8, Shenzhen Mindray Animal Medical Technology Co., Ltd., Shenzhen, China). A two-dimensional short-axis view, followed by M-mode tracings, were used to measure the parameters, including LVPWd, LVPWs, LVIDd, LVIDs, IVSd, and IVSs. Additionally, EDV, ESV, and SV were calculated from three consecutive cardiac cycles. LV fractional shortening was derived using the following formula: %FS = [(LVIDd − LVIDs)/LVIDd] × 100.

### 4.5. Blood Pressure and Heart Rate Measurement in Unconscious Rats

After evaluating cardiac function, SBP, DBP, MAP, PP, and HR were monitored while the rats remained under anesthesia. A polyethylene catheter was inserted into the left femoral artery and connected to a pressure transducer. Blood pressure parameters were continuously recorded for 20 min using Acknowledge Data Acquisition software (Biopac Systems Inc., Santa Barbara, CA, USA). Following this, blood samples were drawn from the abdominal aorta and the thoracic aorta, along with the heart, and were rapidly excised for further analyses.

### 4.6. Vascular Function Study

After hemodynamic measurements were captured, the mesenteric vascular bed was excised and positioned on a stainless-steel grid inside a humidified chamber at a temperature of 35–37 °C. The preparation was continuously perfused with Krebs solution (composition: 118 mM NaCl, 25 mM NaHCO_3_, 4.8 mM KCl, 1.2 mM KH_2_PO_4_, 1.2 mM MgSO_4_·7H_2_O, 1.25 mM CaCl_2_, and 11.1 mM glucose; pH 7.4), and aerated with a 95% O_2_ and 5% CO_2_ gas mixture. The solution was delivered at a constant flow rate of 5 mL/min using a peristaltic pump (07534–04, Cole-Palmer Instrument, Illinois, USA). The procedures followed the methodology described by Pakdeechote et al. (2007) [39]. Sympathetic nerve activity was assessed using electrical field stimulation (EFS; 5–40 Hz, 90 V, 1 ms, for 30 s at 5-min intervals) and by administering NE in increasing concentrations (0.15–15 nmol). To evaluate vasorelaxation, vascular tone was elevated with methoxamine (5–7 μM), followed by the administration of ACh (1 pmol–0.01 μmol) to test endothelium-dependent relaxation or SNP (1 pmol–0.01 μmol) to assess vascular smooth muscle cell relaxation. The vascular responses were recorded as changes in mean perfusion pressure (mmHg) with a pressure transducer connected to a BIOPAC System (BIOPAC Systems Inc., Santa Barbara, CA, USA).

In a separate set of experiments, thoracic aorta segments (2–3 mm in length) were quickly excised and prepared to evaluate their responses to vasoactive agents. Each aortic ring was suspended in a 15 mL organ bath filled with Krebs solution, maintained at 37 °C, and aerated with a gas mixture of 95% O_2_ and 5% CO_2_. Following preconstriction with phenylephrine (10 µM), ACh (0.01–3 µM) and SNP (0.01–3 µM) were added into the organ bath, and the vascular responses were recorded. The resting tension of 1 g was set, and changes in tension were measured using a transducer connected to a 4-channel bridge amplifier, a PowerLab A/D converter, and a computer running Chart v5 software (PowerLab System, ADInstruments, Bella Vista, Australia). Relaxation responses were expressed as a percentage of the contraction induced by phenylephrine.

### 4.7. Histological Study of Aorta and Heart

The excised heart and aorta were promptly fixed in 4% paraformaldehyde, and the tissues were processed following previously described methods [40]. After paraffin embedding, the tissues were sectioned into 5 μm slices and stained with hematoxylin and eosin (H&E) (Bio-Optica Milano S.p.A., Milan, Italy). The alterations in structure were observed using an Eclipse Ni–U upright microscope (Nikon, Tokyo, Japan) at magnifications of ×4 or ×20. Quantitative analyses were conducted and adapted from the methodologies outlined in earlier studies [40,41].

### 4.8. RAS Measurement

#### 4.8.1. Assay of Angiotensin-Converting Enzyme (ACE) Activity

Serum ACE activity was measured using the OPA-chromogenic reaction for histidyl-leucine, based on the method of a published study [42].

#### 4.8.2. Assay of Plasma Ang II Level

The plasma concentration of Ang II was determined using an Ang II enzyme immunoassay kit (RAB0010–1KT, St. Louis, MO, USA). The assay was performed according to the manufacturer’s instructions.

### 4.9. Protein Expression Measurement

Western blot analysis was conducted to assess the protein expression of AT1 receptor and the gp91^phox^ subunit of NADPH oxidase in both aorta and cardiac tissue, following the method described in previous study [43]. In brief, the tissue samples were homogenized, and the proteins were extracted. Sodium dodecyl sulfate-polyacrylamide gel electrophoresis (SDS-PAGE) was used for protein separation, followed by transfer to polyvinylidene difluoride (PVDF) membranes. The membranes were blocked with 5% bovine serum albumin (BSA) in Tris-buffered saline with 0.1% Tween 20 (TBST) for 2 h at room temperature. Primary antibody incubation was carried out overnight at 4 °C with antibody targeting AT1 receptor (G-3) (SC-515884, 1:1000) or gp91^phox^ (G-1) (sc74514, 1:1000). After incubation, TBST was used to wash the membranes three times, and horseradish peroxidase-conjugated secondary antibodies were applied at room temperature. Protein detection was performed using an ECL detection system (Amersham™ ECL™ Prime, Amersham Biosciences Corp., Piscataway, NJ, USA), and visualization was conducted using an Amersham Imager 600 (GE Healthcare Life Sciences, Uppsala, Sweden). The band intensities were quantified and normalized to β-actin, and values were expressed as a percentage relative to the control group from the same gel.

### 4.10. Plasma Nitric Oxide Metabolites (NOx) Measurement

The measurement of the nitric oxide metabolites (nitrate/nitrite) was carried out according to the instructions provided in the nitric oxide assay kit (MAK454, Merck KGaA, Darmstadt, Germany).

### 4.11. Oxidative Stress Marker Measurement

#### 4.11.1. Superoxide Production Measurement

Superoxide production in cardiac and aortic tissue was assessed using the lucigenin-enhanced chemiluminescence method, as described by Lu et al. (1996) [44]. The results were expressed as relative light unit counts per minute per milligram of dry tissue weight.

#### 4.11.2. Malondialdehyde Level Measurement

Plasma and cardiac tissue MDA were determined using the thiobarbituric acid reactive substances (TBARS) assay, a spectrophotometric method outlined in previous studies [45,46]. The levels of plasma MDA were reported in μM, and the levels of tissue MDA in μmol/g protein.

#### 4.11.3. Catalase Activity Measurement

To measure catalase activity in cardiac tissue, a modified version of the method of the previous study was employed [47]. Briefly, cardiac tissue supernatant or standard catalase solution was added to a microplate, and 50 μL of 30% H_2_O_2_ in 50 nM potassium phosphate buffer with pH 7.0 was added to initiate the reaction. The reaction was stopped with the addition of 25 μL of 5 N H_2_SO_4_, followed by 150 μL of KMnO_4_. The absorbance was read at 490 nm, and the catalase activity was expressed as U/g of protein.

Serum catalase activity was determined using the method described in previous studies [48,49]. In this method, 20 μL of serum was mixed with 100 μL of substrate solution (65 μmol/mL H_2_O_2_ in 60 mmol/L sodium-potassium phosphate buffer, pH 7.4) and incubated at 37 °C for 1 min. A total of 100 μL of 32.4 mmol/L ammonium molybdate ((NH_4_)6Mo7O_24_·4H_2_O) was added to stop the reaction. The resulting yellow complex was quantified by measuring the absorbance at 405 nm, and the serum catalase activity was reported in kU/L.

### 4.12. Statistical Analysis

The results are presented as mean ± standard error of the mean (SEM). Comparisons between groups were made using one-way ANOVA, followed by Tukey’s post hoc tests. Statistical significance was set at a p-value of less than 0.05.

## 5. Conclusions

In conclusion, HLO possesses a total phenolic content that demonstrates cardiovascular-protective effects against NOS inhibitor-induced hypertension. HLO exhibits an ACE inhibitory action and inhibits the Ang II/AT1 receptor/NOX2 pathway, alleviating cardiovascular hypertrophy and oxidative stress in a hypertensive rat model. Our findings suggest that HLO displays beneficial effects under a hypertensive condition. However, further research, including the possibility of human trials, is essential to comprehensively assess the safety and effectiveness of HLO in humans. In addition, exploring its underlying molecular mechanisms and assessing its potential as an adjunct to standard antihypertensive therapy could enhance its therapeutic value and support the development of novel approaches for hypertension management.

## Figures and Tables

**Figure 1 ijms-26-01897-f001:**
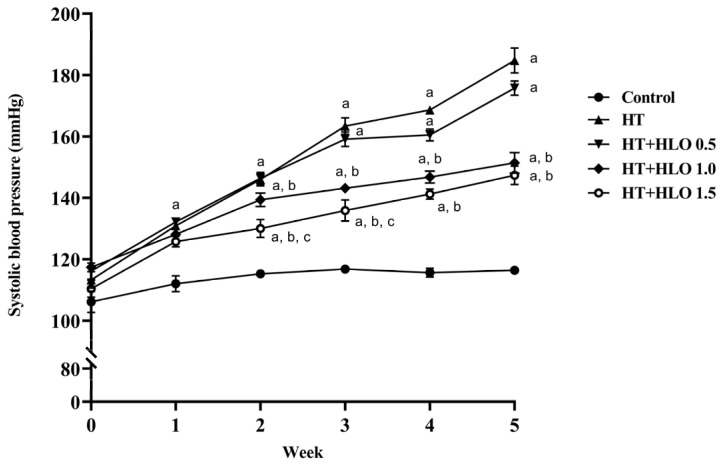
Effect of HLO on weekly SP measured by tail-cuff method. Data are presented as mean ± SEM (*n* = 8/group). ^a^ *p* < 0.05 vs. control, ^b^ *p* < 0.05 vs. HT, ^c^ *p* < 0.05 vs. HT + HLO 0.5; Control, normotensive rats; HT, hypertensive rats; HT + HLO 0.5, hypertensive rats receiving 0.5 mL/kg hemp leaf oil; HT + HLO 1.0, hypertensive rats receiving 1 mL/kg hemp leaf oil; HT + HLO 1.5, hypertensive rats receiving 1.5 mL/kg hemp leaf oil.

**Figure 2 ijms-26-01897-f002:**
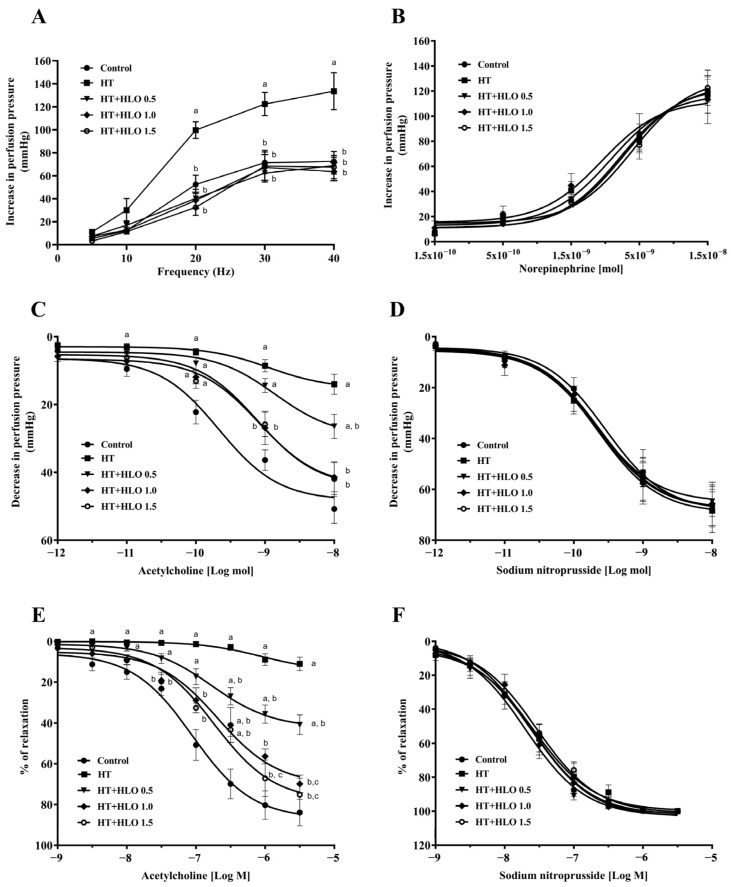
The response to EFS (**A**), NE (**B**), ACh (**C**), and SNP (**D**) in mesenteric vascular beds, as well as the response to ACh (**E**) and SNP (**F**) in aortic rings. Data are presented as mean ± SEM (*n* = 6/group). ^a^ *p* < 0.05 vs. control, ^b^ *p* < 0.05 vs. HT, ^c^ *p* < 0.05 vs. HT + HLO 0.5; Control, normotensive rats; HT, hypertensive rats; HT + HLO 0.5, hypertensive rats receiving 0.5 mL/kg hemp leaf oil; HT + HLO 1.0, hypertensive rats receiving 1 mL/kg hemp leaf oil; HT + HLO 1.5, hypertensive rats receiving 1.5 mL/kg hemp leaf oil.

**Figure 3 ijms-26-01897-f003:**
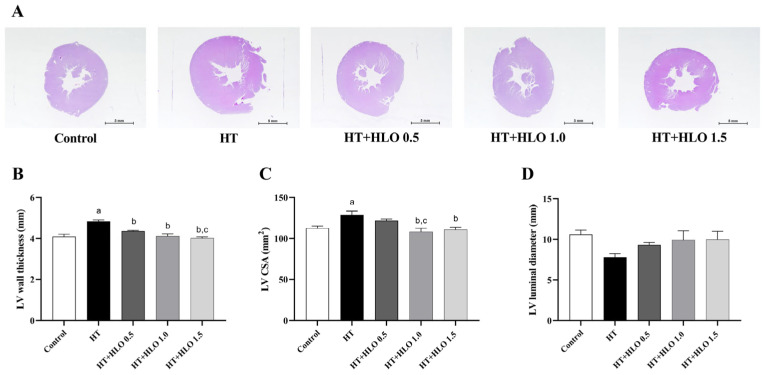
Effect of HLO on LV morphology. The top panel shows the representative H&E staining images from each group (**A**) (magnification × 4, scale bar = 5 mm). The quantitative analysis of the LV wall thickness (**B**), cross-sectional area (CSA) (**C**), and luminal diameter (**D**) are also presented. Data are presented as mean ± SEM (*n* = 6/group). ^a^ *p* < 0.05 vs. control, ^b^ *p* < 0.05 vs. HT, ^c^ *p* < 0.05 vs. HT + HLO 0.5; Control, normotensive rats; HT, hypertensive rats; HT + HLO 0.5, hypertensive rats receiving 0.5 mL/kg hemp leaf oil; HT + HLO 1.0, hypertensive rats receiving 1 mL/kg hemp leaf oil; HT + HLO 1.5, hypertensive rats receiving 1.5 mL/kg hemp leaf oil.

**Figure 4 ijms-26-01897-f004:**
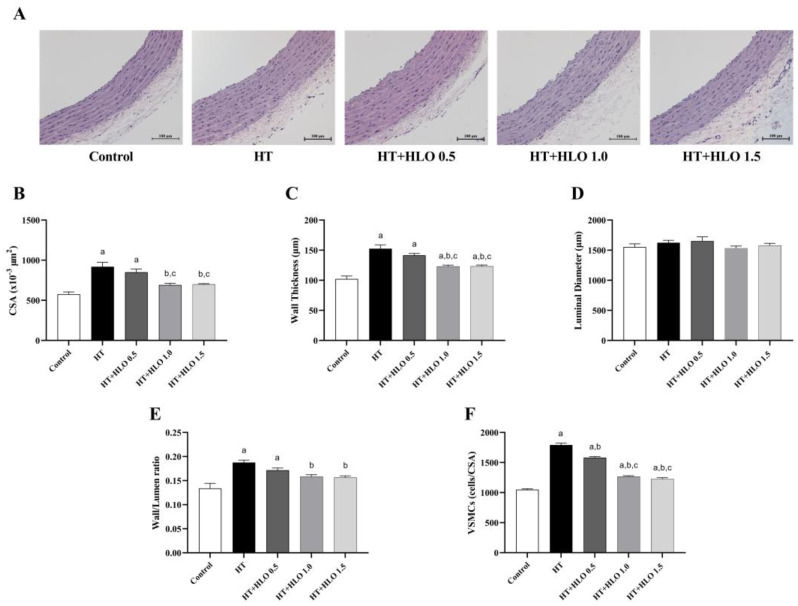
Effect of HLO on aortic morphology. The top panel presents representative H&E staining images for each group (**A**) (magnification × 20, scale bar = 100 μm). The quantitative analysis includes measurement of the aortic cross-sectional area (**B**), wall thickness (**C**), luminal diameter (**D**), wall to lumen ratio (**E**), and vascular smooth muscle cells (**F**). Data are presented as mean ± SEM (*n* = 6/group). ^a^ *p* < 0.05 vs. control, ^b^ *p* < 0.05 vs. HT, ^c^ *p* < 0.05 vs. HT + HLO 0.5; Control, normotensive rats; HT, hypertensive rats; HT + HLO 0.5, hypertensive rats receiving 0.5 mL/kg hemp leaf oil; HT + HLO 1.0, hypertensive rats receiving 1 mL/kg hemp leaf oil; HT + HLO 1.5, hypertensive rats receiving 1.5 mL/kg hemp leaf oil.

**Figure 5 ijms-26-01897-f005:**
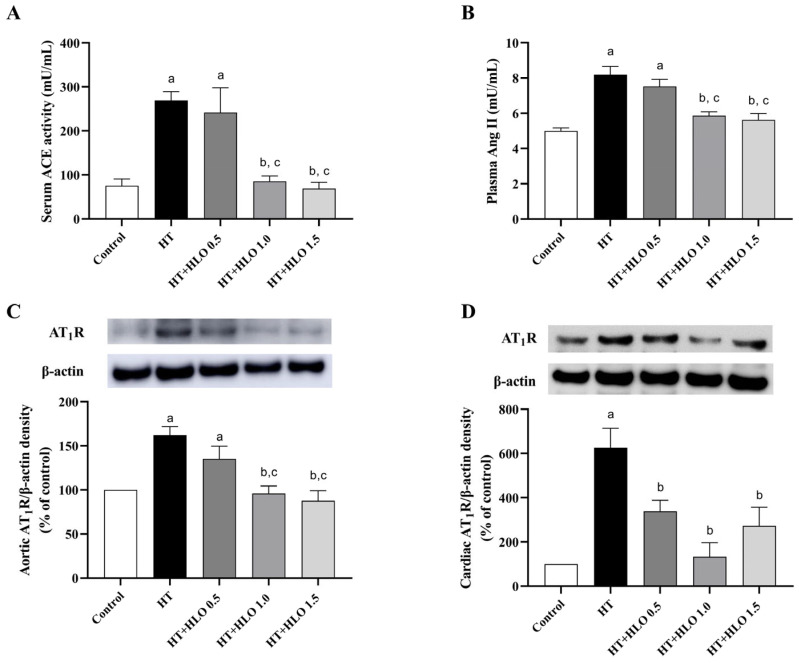
Effect of HLO on serum ACE activity (**A**), plasma Ang II levels (**B**), and AT1 receptor protein expression in aorta (**C**) and cardiac tissues (**D**) in hypertensive rats. Data are presented as mean ± SEM (*n* = 4–6/group). ^a^ *p* < 0.05 vs. control, ^b^ *p* < 0.05 vs. HT, ^c^ *p* < 0.05 vs. HT + HLO 0.5; Control, normotensive rats; HT, hypertensive rats; HT + HLO 0.5, hypertensive rats receiving 0.5 mL/kg hemp leaf oil; HT + HLO 1.0, hypertensive rats receiving 1 mL/kg hemp leaf oil; HT + HLO 1.5, hypertensive rats receiving 1.5 mL/kg hemp leaf oil.

**Figure 6 ijms-26-01897-f006:**
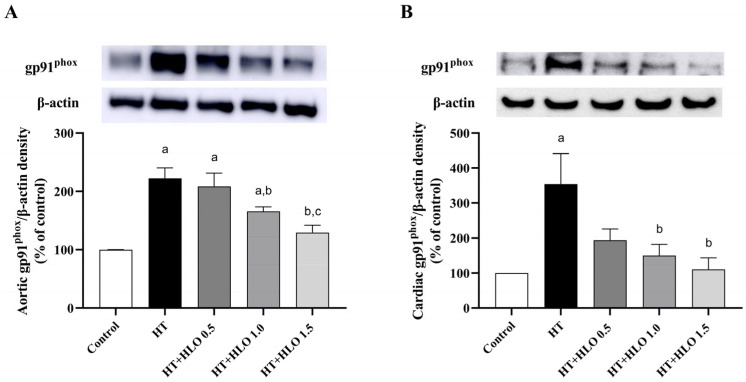
Effect of HLO on gp91^phox^ protein expression in aorta (**A**) and cardiac tissues (**B**) in hypertensive rats. Data are presented as mean ± SEM (*n* = 4–5/group). ^a^ *p* < 0.05 vs. control, ^b^ *p* < 0.05 vs. HT, ^c^ *p* < 0.05 vs. HT + HLO 0.5; Control, normotensive rats; HT, hypertensive rats; HT + HLO 0.5, hypertensive rats receiving 0.5 mL/kg hemp leaf oil; HT + HLO 1.0, hypertensive rats receiving 1 mL/kg hemp leaf oil; HT + HLO 1.5, hypertensive rats receiving 1.5 mL/kg hemp leaf oil.

**Table 1 ijms-26-01897-t001:** Phenolic compounds present in HLO.

Parameters	Concentration
Total phenolic compounds (μg * QE/mL)	26.02 ± 1.14
Total flavonoid contents (μg * QE/mL)	2.10 ± 0.48
Quercetin (mg/100 mL)	23.07 ± 0.45

Data are presented as mean ± SD. * micrograms of quercetin equivalent per mL.

**Table 2 ijms-26-01897-t002:** Effect of HLO on direct blood pressure and heart rate.

Parameter	Control	HT	HT + HLO 0.5	HT + HLO 1.0	HT + HLO 1.5
SBP (mmHg)	115.89 ± 2.50	181.89 ± 4.57 ^a^	158.02 ± 6.38 ^a,b^	137.20 ± 5.09 ^b,c^	134.18 ± 5.90 ^b,c^
DBP (mmHg)	70.58 ± 2.76	128.18 ± 2.37 ^a^	107.25 ± 7.23 ^a,b^	89.14 ± 8.24 ^a,b^	90.20 ± 5.44 ^b^
MAP (mmHg)	85.68 ± 2.37	146.08 ± 2.66 ^a^	124.17 ± 6.69 ^a,b^	105.16 ± 6.46 ^a,b^	104.86 ± 5.54 ^b,c^
PP (mmHg)	45.32 ± 2.62	53.71 ± 4.06	50.77 ± 4.08	48.07 ± 7.40	43.97 ± 1.73
HR (beat/min)	354.70 ± 11.88	373.65 ± 18.38	390.52 ± 8.89	379.40 ± 14.14	372.77 ± 12.33

Data are presented as mean ± SEM (*n* = 8/group). ^a^ *p* < 0.05 vs. control, ^b^ *p* < 0.05 vs. HT, ^c^ *p* < 0.05 vs. HT + HLO 0.5; SBP, systolic blood pressure; DBP, diastolic blood pressure; MAP, mean arterial pressure; PP, pulse pressure; HR, heart rate; Control, normotensive rats; HT, hypertensive rats; HT + HLO 0.5, hypertensive rats receiving 0.5 mL/kg hemp leaf oil; HT + HLO 1.0, hypertensive rats receiving 1 mL/kg hemp leaf oil; HT + HLO 1.5, hypertensive rats receiving 1.5 mL/kg hemp leaf oil.

**Table 3 ijms-26-01897-t003:** Effect of HLO on body weight and organ weight.

Parameter	Control	HT	HT + HLO 0.5	HT + HLO 1.0	HT + HLO 1.5
BW (g)	474.00 ± 11.19	445.12 ± 6.77	486.25 ± 11.15	486.5 ± 10.49	478.00 ± 6.49
HW (g)	1.38 ± 0.04	1.54 ± 0.04	1.59 ± 0.10	1.53 ± 0.07	1.62 ± 0.03 ^a^
HW/BW (mg/g)	2.92 ± 0.13	3.47 ± 0.11 ^a^	3.25 ± 0.15	3.15 ± 0.17	3.40 ± 0.03
LVW (g)	0.85 ± 0.02	1.02 ± 0.04 ^a^	0.95 ± 0.03	0.95 ± 0.03	0.95 ± 0.01
LVW/BW (mg/g)	1.80 ± 0.03	2.29 ± 0.10 ^a^	1.95 ± 0.03 ^b^	1.94 ± 0.05 ^b^	1.98 ± 0.03 ^b^

Data are presented as mean ± SEM (*n* = 6/group). ^a^ *p* < 0.05 vs. control, ^b^ *p* < 0.05 vs. L-NAME; BW, body weight; HW, heart weight; LVW, left ventricular weight; Control, normotensive rats; HT, hypertensive rats; HT + HLO 0.5, hypertensive rats receiving 0.5 mL/kg hemp leaf oil; HT + HLO 1.0, hypertensive rats receiving 1 mL/kg hemp leaf oil; HT + HLO 1.5, hypertensive rats receiving 1.5 mL/kg hemp leaf oil.

**Table 4 ijms-26-01897-t004:** Effect of HLO on echocardiograph parameters.

Parameters	Control	HT	HT + HLO 0.5	HT + HLO 1.0	HT + HLO 1.5
LVPWd (mm)	1.79 ± 0.06	2.13 ± 0.13 ^a^	1.92 ± 0.07	1.75 ± 0.05 ^b^	1.76 ± 0.04 ^b^
LVPWs (mm)	2.66 ± 0.10	2.85 ± 0.14	2.73 ± 0.12	2.64 ± 0.10	2.74 ± 0.12
LVIDd (mm)	7.39 ± 0.13	5.96 ± 0.40 ^a^	6.73 ± 0.16	6.73 ± 0.16	6.83 ± 0.10
LVIDs (mm)	4.13 ± 0.16	3.94 ± 0.26	4.00 ± 0.19	3.64 ± 0.20	3.7 ± 0.21
IVSd (mm)	1.66 ± 0.04	1.93 ± 0.10 ^a^	1.90 ± 0.06	1.72 ± 0.04	1.76 ± 0.04
IVSs (mm)	2.55 ± 0.18	2.79 ± 0.09	2.74 ± 0.12	2.57 ± 0.12	2.84 ± 0.13
EDV (mL)	0.9 ± 0.04	0.53 ± 0.09 ^a^	0.70 ± 0.04	0.70 ± 0.05	0.73 ± 0.03
ESV (mL)	0.17 ± 0.02	0.16 ± 0.03	0.17 ± 0.02	0.13 ± 0.02	0.13 ± 0.02
SV (mL)	0.73 ± 0.04	0.36 ± 0.06 ^a^	0.53 ± 0.04 ^a^	0.58 ± 0.05 ^b^	0.59 ± 0.02 ^b^
EF (%)	80.74 ± 1.89	67.6 ± 3.14 ^a^	75.83 ± 2.98	81.5 ± 2.50 ^b^	81.67 ± 2.20 ^b^
FS (%)	44.63 ± 1.92	33.37 ± 2.39 ^a^	40.32 ± 2.56	46.02 ± 2.96 ^b^	45.96 ± 2.58 ^b^

Data are presented as mean ± SEM (*n* = 6/group). ^a^ *p* < 0.05 vs. control, ^b^ *p* < 0.05 vs. HT; LVPWd, left ventricular posterior wall thickness during diastole; LVPWs, left ventricular posterior wall thickness during systole; LVIDd, left ventricular internal dimension at end-diastole; LVIDs, left ventricular internal dimension at end-systole; IVSd, interventricular septum during diastole; IVSs, interventricular septum during systole; EDV, end-diastolic volume; ESV, end-systolic volume; SV, stroke volume; EF, ejection fraction; FS, fractional shortening; Control, normotensive rats; HT, hypertensive rats; HT+HLO 0.5, hypertensive rats receiving 0.5 mL/kg hemp leaf oil; HT+HLO 1.0, hypertensive rats receiving 1 mL/kg hemp leaf oil; HT + HLO 1.5, hypertensive rats receiving 1.5 mL/kg hemp leaf oil.

**Table 5 ijms-26-01897-t005:** Level of NOx and oxidative stress parameters.

Parameter	Control	HT	HT + HLO 0.5	HT + HLO 1.0	HT + HLO 1.5
Plasma NOx (µM)	92.96 ± 11.79	56.51 ± 8.27 ^a^	67.90 ± 7.70	89.92 ± 16.95 ^b^	87.73 ± 9.18 ^b^
Superoxide production in aorta (Count/mg dry wt/min)	76.08 ± 5.99	160.46 ± 26.03 ^a^	110.93 ± 20.41^a^	80.76 ± 7.68 ^b^	60.14 ± 9.84 ^b^
Superoxide production in cardiac tissues (Count/mg dry wt/min)	60.69 ± 12.33	150.98 ± 41.01^a^	118.78 ± 11.34	67.92 ± 15.92 ^b^	63.31 ± 12.93 ^b^
Plasma MDA (µM)	7.08 ± 0.69	25.65 ± 6.52 ^a^	20.36 ± 6.01	8.89 ± 1.59 ^b^	9.88 ± 1.52 ^b^
Cardiac tissues MDA (µmol/g protein)	3.98 ± 1.46	19.03 ± 2.58 ^a^	16.27 ± 2.55 ^a^	4.16 ± 1.15 ^b,c^	3.07 ± 0.87 ^b,c^
Serum catalase (kU/L)	270.92 ± 14.01	148.97 ± 20.82 ^a^	202.54 ± 11.02 ^a^	243.78 ± 7.78 ^b^	250.60 ± 20.39 ^b^
Cardiac tissues catalase (U/g protein)	15.88 ± 0.95	8.03 ± 1.46 ^a^	9.39 ± 0.77	16.73 ± 2.33 ^b,c^	17.61 ± 2.13 ^b,c^

Data are presented as mean ± SEM (*n* = 6–8/group). ^a^ *p* < 0.05 vs. control, ^b^ *p* < 0.05 vs. HT, ^c^ *p* < 0.05 vs. HT + HLO 0.5; Control, normotensive rats; HT, hypertensive rats; HT + HLO 0.5, hypertensive rats receiving 0.5 mL/kg hemp leaf oil; HT + HLO 1.0, hypertensive rats receiving 1 mL/kg hemp leaf oil; HT + HLO 1.5, hypertensive rats receiving 1.5 mL/kg hemp leaf oil.

## Data Availability

The datasets used and analyzed during the current study are available from the corresponding author on reasonable request.

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
