# Peer review of "Cannabis sativa L. Leaf Oil Displays Cardiovascular Protective Effects in Hypertensive Rats"

_ijms, 2025, doi:10.3390/ijms26051897_

Round 1

Reviewer 1 Report

Comments and Suggestions for Authors

Review of the manuscript entitled: Cannabis sativa L. leaf oil displays cardiovascular protective effects in hypertensive rats. The manuscript prepared by the authors concerns one topic of the use of Cannabis sativa in medicine. A major deficiency in the manuscript is the lack of explanation of the molecular mechanism. On the other hand, the positive side is the fact that studies were performed on animals. In my opinion some corrections should be made before publication.

1.      I believe that the abstract and introduction are prepared correctly. However, the aim of the manuscript should be clearly indicated at the end of introduction e.g. "The aim of the present study was to ...". A clear purpose makes it easier to understand the intentions of Authors. Furthermore it is not acceptable to present your findings in the introduction, do the Authors know the difference between an introduction and an abstract? It is not the same. For this reason the last paragraph of the introduction should be corrected.

2.      In the description of figures and tables the "n" value is missing, what was the number of repetitions?

3.      Figure 4 panel E missing unit on axis, check other figures. Units must be marked on each axis.

4.      In my opinion the discussion is written quite well. However, the conclusions should be highlighted in a separate section according to the journal's guidelines (5. Conclusions). In addition, medical consequences and their in-depth analysis should be included in the conclusions

5.      If possible exact location (latitude and longitude) should be added line 314.

6.      Detailed chromatographic analysis should be added to the manuscript.

Author Response

Comment 1: I believe that the abstract and introduction are prepared correctly. However, the aim of the manuscript should be clearly indicated at the end of introduction e.g. "The aim of the present study was to ...". A clear purpose makes it easier to understand the intentions of Authors. Furthermore it is not acceptable to present your findings in the introduction, do the Authors know the difference between an introduction and an abstract? It is not the same. For this reason the last paragraph of the introduction should be corrected.

Response 1: We revised the last paragraph of the introduction as reviewer suggested as follow: Due to the limited information available on the impact of HLO on NO-deficient hypertension, particularly concerning cardiovascular alterations, the aim of the present study was to investigate the effect of HLO on blood pressure and cardiovascular changes in Sprague-Dawley rats treated with L-NAME. Additionally, we aimed to investigate the possible mechanisms underlying of these effects.

Comment 2: In the description of figures and tables the "n" value is missing, what was the number of repetitions?

Response 2: The n value was added in all figure and table legends as highlighted.

Comment 3: Figure 4 panel E missing unit on axis, check other figures. Units must be marked on each axis.

Response 3: Figure 4E is the ratio of the comparison of two quantities using division (Outer wall diameter/Lumen diameter). It expressed in numbers in its simplest form, and has no unit.

Comment 4: In my opinion the discussion is written quite well. However, the conclusions should be highlighted in a separate section according to the journal's guidelines (5. Conclusions). In addition, medical consequences and their in-depth analysis should be included in the conclusions.

Response 4: The part 5. Conclusions was added and revised as reviewer suggested as follow: In conclusion, HLO possesses total phenolic content that demonstrates cardio-vascular-protective effects against NOS inhibitor-induced hypertension. HLO exhibits an ACE inhibitory action and inhibits the Ang II/AT1 receptor/NOX2 pathway, alleviating cardiovascular hypertrophy and oxidative stress in a hypertensive rat model. Our findings suggest that HLO has beneficial effects in hypertensive condition. How-ever, further research, including the possibility of human trials, is essential to comprehensively assess the safety and effectiveness of HLO in humans. In addition, exploring its underlying molecular mechanisms and assessing its potential as an adjunct to standard antihypertensive therapy could enhance its therapeutic value and support the development of novel approaches for hypertension management.

Comment 5:  If possible exact location (latitude and longitude) should be added line 314.

Response 5: The latitude and longitude of the location were added as highlighted.

Comment 6: Detailed chromatographic analysis should be added to the manuscript.

Response 6: The detailed of chromatographic analysis has been added in section 4.1 as follow: “The quantification of cannabinoids in hemp leaf was measured by HPLC method using a Shimadzu Prominence-i LC-2030 C 3D Plus system, equipped with a Millipore Sigma Ascentis-C18 Express column (2.7 μm x 150 mm x 3 mm) and a photodiode ar-ray detector (PDA). The column was operated with solvent A as HPLC water, 8% (v/v) methanol, 0.035% (v/v) formic acid, and 1.8 mM ammonium formate, and solvent B as HPLC acetonitrile, with a flow rate of 0.45 mL/min, the UV detector was set at 220 nm.”

Reviewer 2 Report

Comments and Suggestions for Authors

The study investigates the cardioprotective effect of hemp leaf oil in nitric oxide synthase inhibition-induced hypertensive rats. The study showed that five-week administration of HLO reduced the elevation of blood pressure, improved cardiac function and reduced myocardial hypertrophy, improved vascular function by reducing contraction induced by the sympathetic nervous system and improving endothelial relaxation, and inhibited the activation of the renin-angiotensin system (RAS) and oxidative stress via the Ang II/AT1 receptor/NOX2 pathway.

The research is pertinent, in my view, since it determines a possible natural therapy with antihypertensive and cardioprotective properties. Elucidation of the mechanisms through which HLO operates can ultimately result in the creation of new hypertension therapies, particularly for nitric oxide imbalance and heightened oxidative stress-associated cases.

The paper can be publishable, in my view, subject to addressing some major concerns.

-The introduction presents useful information about the pharmacological effects of cannabis and how nitric oxide is relevant to blood pressure regulation. Some points, however, are redundant or indirectly related to the final objective of the study. The introduction would fare better if it focused more on the interconnection between cannabis leaf oil (HLO) and hypertension, revealing more clearly the scientific gap the study seeks to fill.

-The current research demystifies the intricate mechanisms through which HLO works, with special focus on its role of inhibiting the overactivity of the renin-angiotensin system (RAS) and alleviating oxidative stress. Yet, how these mechanisms are interpreted in the long-term control of hypertension is yet to be explained.

-The findings are well presented in tables and figures; however, the accompanying descriptions are in some areas too superficial and do not include sufficient highlighting of the importance of the statistical variations. In some areas, the comparisons of study groups appear to be restated without indicating clearly what findings are pivotal to drawing conclusions. A summary of the key findings presented in the text along with the tables would be useful so that the reader can pinpoint the key differences and their importance.

-The paper makes a very firm conclusion about the potential therapeutic application of HLO in the management of hypertension; it does not, however, have sufficient detail on the way forward for further research. A section outlining potential clinical implications, such as progression to human trials or examination of the effect of HLO as an adjunct to standard antihypertensive therapy, would greatly increase the value of the findings.

Author Response

Comment 1: The introduction presents useful information about the pharmacological effects of cannabis and how nitric oxide is relevant to blood pressure regulation. Some points, however, are redundant or indirectly related to the final objective of the study. The introduction would fare better if it focused more on the interconnection between cannabis leaf oil (HLO) and hypertension, revealing more clearly the scientific gap the study seeks to fill.

Response 1: We have revised the introduction to more clearly highlight the research gap as follow: Cannabis sativa L. and its bioactive compounds have been increasingly studied for their potential cardiovascular effects. However, its mechanism of action is complex and has multidirectional influence on the cardiovascular system. Nevertheless, limited re-search exists regarding the blood pressure and cardiovascular effects of hemp leaf oil (HLO).

Comment 2: The current research demystifies the intricate mechanisms through which HLO works, with special focus on its role of inhibiting the overactivity of the renin-angiotensin system (RAS) and alleviating oxidative stress. Yet, how these mechanisms are interpreted in the long-term control of hypertension is yet to be explained.

Response 2: We added the mechanism of HLO in the long-term control of hypertension that associated with overactivity of the renin-angiotensin system (RAS) and alleviating oxidative stress in discussion section as follow: It is well established that RAS overactivity and oxidative stress play crucial roles in the long-term regulation of high blood pressure in L-NAME treated rats [32]. Thus, the role of HLO in the long-term regulation of blood pressure, particularly in relation to RAS activation and oxidative stress, has been revealed.

Reference: Rincón, J.; Correia, D.; Arcaya, J.L.; Finol, E.; Fernández, A.; Pérez, M.; Yaguas, K.; Talavera, E.; Chávez, M.; Summer, R.; et al. Role of Angiotensin II type 1 receptor on renal NAD(P)H oxidase, oxidative stress and inflammation in nitric oxide inhibition induced-hypertension. Life Sciences 2015, 124, 81-90, doi:https://doi.org/10.1016/j.lfs.2015.01.005.

Comment 3: The findings are well presented in tables and figures; however, the accompanying descriptions are in some areas too superficial and do not include sufficient highlighting of the importance of the statistical variations. In some areas, the comparisons of study groups appear to be restated without indicating clearly what findings are pivotal to drawing conclusions. A summary of the key findings presented in the text along with the tables would be useful so that the reader can pinpoint the key differences and their importance.

Response 3: We have included a summary in the text that integrates the key findings of the results as highlighted.

Comment 4: The paper makes a very firm conclusion about the potential therapeutic application of HLO in the management of hypertension; it does not, however, have sufficient detail on the way forward for further research. A section outlining potential clinical implications, such as progression to human trials or examination of the effect of HLO as an adjunct to standard antihypertensive therapy, would greatly increase the value of the findings.

Response 4: We have revised the explanation of clinical implications as reviewer suggested, in the conclusion section as follow: Our findings suggest that HLO has beneficial effects in hypertensive condition. However, further research, including the possibility of human trials, is essential to comprehensively assess the safety and effectiveness of HLO in humans. In addition, exploring its underlying molecular mechanisms and assessing its potential as an adjunct to standard antihypertensive therapy could enhance its therapeutic value and support the development of novel approaches for hypertension management.    

Round 2

Reviewer 2 Report

Comments and Suggestions for Authors

The authors have revised the manuscript in accordance with the suggestions provided. I recommend the publication of the article.